SCIENCE FORUM

# An open investigation of the reproducibility of cancer biology research

**Abstract** It is widely believed that research that builds upon previously published findings has reproduced the original work. However, it is rare for researchers to perform or publish direct replications of existing results. The Reproducibility Project: Cancer Biology is an open investigation of reproducibility in preclinical cancer biology research. We have identified 50 high impact cancer biology articles published in the period 2010-2012, and plan to replicate a subset of experimental results from each article. A Registered Report detailing the proposed experimental designs and protocols for each subset of experiments will be peer reviewed and published prior to data collection. The results of these experiments will then be published in a Replication Study. The resulting open methodology and dataset will provide evidence about the reproducibility of high-impact results, and an opportunity to identify predictors of reproducibility.

**TIMOTHY M ERRINGTON\*, ELIZABETH IORNS, WILLIAM GUNN, FRASER ELISABETH TAN, JOELLE LOMAX AND BRIAN A NOSEK\***

**\*For correspondence:** tim@cos.io (TME); nosek@virginia.edu (BAN)

**Reviewing editor**: Peter Rodgers, eLife, United Kingdom

Two central features of science are transparency and reproducibility (*Bacon, 1267/1859*; *Jasny et al., 2011*; *Kuhn, 1962*; *Merton, 1942*; *Popper, 1934/1992*). Transparency requires scientists to publish their methodology and data so that the merit of a claim can be assessed on the basis of the evidence rather than the reputation of those making the claim. Reproducibility can refer to both the ability of others to reproduce the findings, given the original data, and to the generation of new data that supports the same conclusions. The focus of this article and project is on the latter—the replication of prior results with new data.

Despite being a defining feature of science, reproducibility is more an assumption than a practice in the present scientific ecosystem (*Collins, 1985*; *Schmidt, 2009*). Incentives for scientific achievement prioritize innovation over replication (*Alberts et al., 2014*; *Nosek, et al., 2012*). Peer review tends to favor manuscripts that contain new findings over those that improve our understanding of a previously published finding.

Moreover, careers are made by producing exciting new results at the frontiers of knowledge, not by verifying prior discoveries.

Reproducing prior results is challenging because of insufficient, incomplete, or inaccurate reporting of methodologies (*Hess, 2011*; *Prinz et al., 2011*; *Steward et al., 2012*; *Hackam and Redelmeier, 2006*; *Landis et al., 2011*). Further, a lack of information about research resources makes it difficult or impossible to determine what was used in a published study (*Vasilevsky et al., 2013*). These challenges are compounded by the lack of funding support available from agencies and foundations to support replication research. When replications are performed, they are rarely published (*Collins, 1985*; *Schmidt, 2009*). A literature review in psychological science, for example, estimated that 0.15% of the published results were direct replications of prior published results (*Makel et al., 2012*). Finally, reproducing analyses with prior data is difficult because researchers are often reluctant to share data, even when required by funding bodies or scientific societies

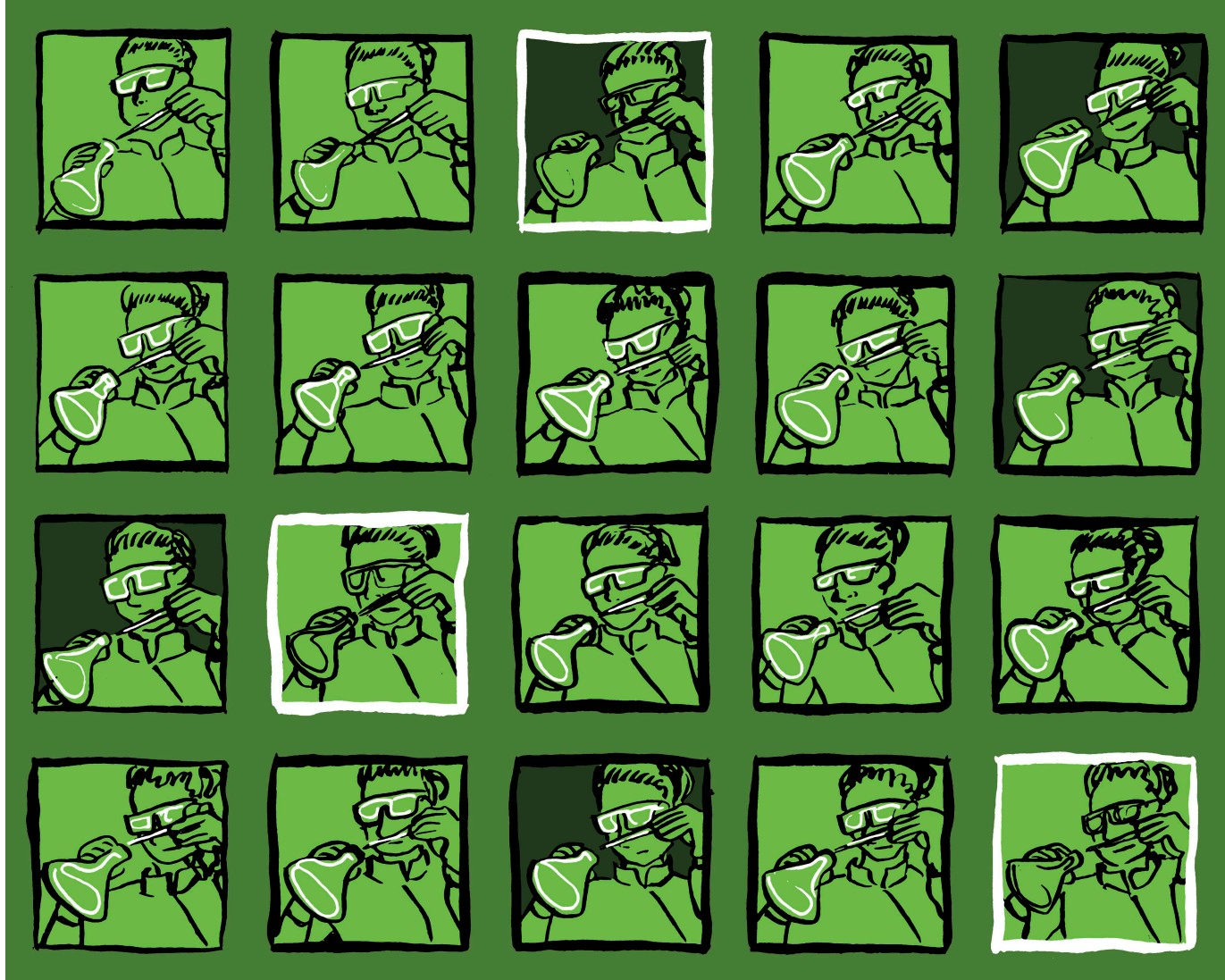

**Figure 1**. The Reproducibility Project: Cancer Biology will replicate selected experiments from a set of 50 research papers in an effort to estimate the rate of reproducibility in preclinical cancer biology research.

ILLUSTRATION: www.claudiastocker.com

(*Wicherts et al., 2006*), and because data loss increases rapidly with time after publication (*Vines et al., 2014*).

If all published results were true and their effect sizes estimated precisely, then a singular focus on innovation over verification might be inconsequential, because the effect size (which is a measure of the strength of the association between variables, or the magnitude of the differences between experimental conditions) would be reliable. In such a context, the most efficient means of knowledge accumulation would be to spend all resources on discovery and trust that each published result provided an accurate estimate of effects on which to build or extend. However, if not all published results are true and if effect sizes are misestimated, then an absence of replication and verification will lead to a published literature that misrepresents reality. The consequences of that scenario would depend on the magnitude of the misestimation.

The accumulating evidence suggests an alarming degree of misestimation. Across disciplines, most published studies demonstrate positive results—results that indicate an expected association between variables or a difference between experimental conditions (*Sterling, 1959*; *Fanelli, 2010*, *2012*). Simultaneously, across disciplines, the average power of studies to detect positive results is quite low (*Cohen, 1962*; *Ioannidis, 2005*; *Button et al., 2013*). In neuroscience, for

example, Button et al. observed the median power of studies to be 21% (**Button et al., 2013**), which means that assuming the finding being investigated is true and accurately estimated, then only 21 of every 100 studies investigating that effect would detect statistically significant evidence for the effect. Most studies would miss detecting the true effect.

The implication of very low power is that the research literature would be filled with lots of negative results, regardless of whether the effects actually exist or not. In the case of neuroscience, assuming all investigated effects in the published literature are true, only 21% of the studies should have obtained a significant, positive result detecting that effect. However, Fanelli observed a positive result rate of 85% in neuroscience (**Fanelli, 2010**). The samples in these two studies were not the same, but both were large and there is little reason to expect lack of comparability. This discrepancy between observed power and observed positive results is not statistically possible. Instead, it suggests systematic exclusion of negative results (**Greenwald, 1975**) and possibly the exaggeration of positive results by employing flexibility in analytic and reporting practices that inflate the likelihood of false positives (**Simmons et al., 2011**).

The small amount of direct evidence about reproducibility converges with the conclusions of these systematic reviews. A survey of faculty and trainees at the MD Anderson Cancer Center found half of those researchers reported an inability to reproduce data on at least one occasion (**Mobley et al., 2013**). More dramatically, two industrial laboratories, Bayer and Amgen, reported reproducibility rates of 11% and 25% in two independent efforts to reproduce findings from dozens of groundbreaking basic science studies in oncology and related areas (**Begley and Ellis, 2011**; **Prinz et al., 2011**).

The available evidence suggests that published research is less reproducible than assumed and desired, perhaps because of an inflation of false positives and a culture of incentives that values publication over accuracy (**Nosek et al., 2012**), but, the evidence is incomplete. The Bayer and Amgen reports of failing to reproduce a high proportion of results provide the most direct evidence. However, neither report made available the effects investigated, the sampling process, the methodology, or the data that comprised the replication efforts (**Nature, 2012**).

It is vitally important to obtain transparent evidence about the reproducibility of scientific research. To that end, this article introduces the Reproducibility Project: Cancer Biology. This project will conduct replications of findings from 50 high-impact articles in the field of cancer biology published between 2010 and 2012. We selected cancer biology as the area of focus because of the Bayer and Amgen reports, and because of the direct importance of efficient progress in this research discipline for the treatment of disease. The project follows a protocol similar to the Reproducibility Project: Psychology (**Open Science Collaboration, 2012**, **2014**), and—in conjunction with *eLife*—adopts an approach in which the proposed experiments and protocols are written up in a Registered Report that is peer reviewed and published prior to data collection (**Chambers et al., 2014**; **Nosek and Lakens, 2014**). Following completion of data collection and analysis for each study, the results of the experiments detailed in the Registered Report are published in a Replication Study.

## The Reproducibility Project: Cancer Biology

Replicating a large number of randomly selected studies is the best approach to obtain an estimate of the rate of reproducibility. However, the current incentive structures that strongly favor innovation over replication mean that it is not in the professional interest of any individual scientist or laboratory to conduct and publish replications, particularly many replications. The Reproducibility Project: Cancer Biology circumvents these barriers by establishing a core team to design, prepare, and monitor project operations, and by spreading the data collection effort across multiple researchers and laboratories.

### Selecting experiments for replication

Resources are finite. Replication is not needed for findings that have no impact, but it can be of substantial value for increasing the confidence and precision of effects that have substantial impact. We identified a sampling frame that balanced breadth of sampling for general inference with sensible investment of resources on replication projects.

The sampling frame was defined as the 400 most cited papers from both Scopus and Web of Science using the search terms (cancer, onco*, tumor*, metasta*, neoplas*, malignan*, carcino*) for 2010, 2011, and 2012. Citations were counted from all sources, which include primary research articles and reviews. This produced an initial sample of 501 articles from 2010, 444 from 2011, and

438 from 2012. Altmetrics scores from Mendeley and Altmetric.com were collected for the entire dataset and used to create a final impact score for each paper. Citation rates and altmetric scores were each standardized by dividing each metric by the highest in the dataset to give each paper a normalized metric score between 0 and 1, which was summed to create an aggregate impact score. Within each year, articles were reviewed for inclusion eligibility starting with the highest aggregate impact article. Articles were removed if they were clinical trials, case studies, reviews, or if they required specialized samples, techniques, or equipment that would be difficult or impossible to obtain. Also, articles reporting sequencing results, such as publications from The Cancer Genome Atlas project, were excluded. However, if sequencing or proteomic experiments were only part of an article, the other experiments in those papers could still be eligible. Review of articles continued until a total of 50 articles, about one-third from each year, were identified as eligible. The final set included 17 papers from 2010, 17 from 2011, and 16 from 2012. From each paper, a subset of experiments were identified for replication, prioritizing those that support the main conclusions of the paper while also attending to feasibility and resource constraints. Details on the selection process and a list of the selected and excluded papers are available at the Open Science Framework.

There are a variety of alternative sampling strategies that could be pursued in parallel efforts such as community nomination of findings that are important to replicate, request from authors to have their published findings replicated (e.g., the Reproducibility Initiative project), or selection of a sample from a particular journal or on a specific topic for focused investigation. The present sampling strategy focuses replication efforts on high impact papers. This could limit the generalizability of inference to all cancer biology research, but has the benefit of increasing precision and attention to the research that is shaping the field.

### Preparing and conducting the replications

The replication experiments are being coordinated by a core group of researchers and conducted by research providers from the Science Exchange network. Because the network consists of over 900 labs skilled in the techniques necessary for replicating the experiments within the chosen studies, the likelihood of a failed replication due to lack of relevant expertise is minimized. The providers are matched to an experiment on

the basis of their skills and available instrumentation, often with multiple providers contributing to each replication. An advantage of these labs - commercial contract research organizations (CROs) and core facilities—is that they are less likely to be biased for or against replicating the effect. This may reduce the effect of experimenter expectations on observed results (*Rosenthal and Fode, 1963*). However, it does not necessarily eliminate expectancy effects as the replicating researchers are aware of the original findings, and they may have expectations about whether the same result is likely to be obtained or not. Another advantage of this approach is that the time and cost of replicating an experiment via Science Exchange is less than that required to establish a collaboration with another academic lab, allowing the project to scale up efficiently.

A community of volunteers, largely composed of postdocs in the life sciences, contributed to the project by extracting information from the original papers and drafting protocols for replication experiments. Information about the project including its coordination, planning, execution, and ultimately the replication data is available publicly at the Open Science Framework. Conducting the project in an open manner increases the accountability, and ideally, the quality of the project and the replications.

A standardized procedure is followed to minimize irrelevant variation between each replication and to maximize the quality of the replication efforts. We aim to conduct the experimental procedure as closely as possible to the original experiment using the same materials and instrumentation, if available. The replication protocol requires the core team to contact the original corresponding author to request materials and any available information that could improve the quality of the replication attempt.

Each replication experiment must have high statistical power (1-ß ≥ 0.80) for observing the original effect size in order to minimize the likelihood of failing to replicate because of low power (i.e., a false negative). However, it is common in biomedical research for some experiments to be presented with representative images or graphs without any inferential testing. In these cases, we will inquire with the original authors if there are additional unpublished replications, if it is not already stated in the article, and if any details are available about the results. Further, these qualitative experiments will be replicated three times and all results will be presented. Because the original representative image presents a mean, but no variability information from sampling, it is

not possible to compute a standardized effect size. Using the mean of the replication to calculate post-hoc power would be invalid for computing the needed effect size, but the variability of the three replications may be a less biased estimator. So, to determine if more than three replications are needed, we will use the original experiment for estimating the mean and the replications for estimating the variability needed to compute power. This strategy will provide an opportunity to identify the need for more sampling than the default of three. As such, it can only result in increasing the overall power of the investigation.

An easy way to fail to replicate a result is to do a terrible job of implementing the Materials and methods or conducting the data analysis. Our priority is to maximize the quality of each replication to adequately test the research question. We do this by conducting the entire project transparently so that error in sampling, design, data collection, and analysis can be identified. Moreover, replications describe in detail the entire experimental design, including controls, conditions, assay optimizations, materials, protocols, and analysis plans prior to initiating data collection. Next, a key part of matching experiments with laboratories is to identify labs with the appropriate expertise to maximize research quality. This is particularly important with new and innovative techniques, though most techniques called for in the selected experiments are standard techniques for which expertise is widely available. As experiments are matched to labs, it is possible that no appropriate service provider can be identified. If appropriate expertise is not available, then the finding or paper will be excluded from the project.

Once the experimental designs and protocols are prepared, the core team solicits feedback from the original authors to identify ways to improve the design. Author input is incorporated into the designs and protocols prior to data collection; suggestions or concerns from the original authors that are not implemented are recorded in the Registered Report. The replication team will also conduct a literature review for evidence of existing replications although, as noted above, direct replication is likely to be rare. Existing published evidence for replication might indicate the likelihood of reproducing the original results.

### Registered Reports

In addition to the informal information exchange between the core team and the original authors, each study will undergo peer review prior to data collection following the Registered Report format (*Chambers et al., 2014*; *Nosek and Lakens, 2014*). Peer reviewers at *eLife*, including subject experts and a statistician, will evaluate the appropriateness and quality of the experimental designs and protocols for replication, as described in the Registered Report. If the Registered Report passes peer review, it will be published prior to data collection. Publication of the Replication Study is then contingent on the replication team following through with the approved design, data collection, and analysis plan; publication of the Replication Study is not contingent on the results. This places the incentives for the replication team and the reviewers on maximizing the quality of methodology and minimizes incentives for achieving a specific result.

The accepted experimental designs and protocols described in the Registered Report will be preregistered publicly at the Open Science Framework and the Registered Report for each study will be published by *eLife* before any experiments are performed. Following completion of data collection and analysis the Replication Study will be published by *eLife* with all data, analysis scripts, reports, and other research materials added to the project on the Open Science Framework for the research community to view, critique, or extend. The collected body of evidence will be the largest public dataset for investigating reproducibility in cancer biology.

## Evaluation of reproducibility

What is a successful replication? A seemingly easy answer is that the replication produces the same result as the original. However, few results are easily categorized as either the same or different. One approach is to consider whether the replication achieves a p-value of less than 0.05 with the same direction of association or ordinal ranking between conditions as the original. This provides some information but still treats experimental outputs dichotomously, with an arbitrary significance threshold. For example, the above approach would classify a replication with p = 0.06 as a failure. An alternative approach in the null hypothesis significance testing framework is to treat the original effect size as the null and test whether the replication is significantly different from that value. This provides complementary information to the first approach because effects can succeed or fail on one or both tests. For example, a replication with p = 0.06 fails the first test but may not be distinguishable from the original on the second test.

Another approach is to compare the effect sizes of the original and replication studies and to then compare whether the estimates are within each other's confidence intervals. This starts to move the inference process away from dichotomous classification and toward estimating effect magnitude and precision of estimation. In another approach, the totality of evidence for an effect is represented by the meta-analytic estimate combining the original and new experiments. This combines all evidence and provides an indication of the present knowledge of the effect.

There is no single answer to the question 'what is a successful replication?' (*Valentine et al., 2011*). As such, we will report multiple indicators of comparison and combination of original and replication effects in order to gain a better understanding of the findings examined and of reproducibility more generally.

### What will and will not be learned

The primary goals of this project are to produce an initial estimate of the reproducibility of cancer biology research and to identify predictors of reproducibility. These are big questions for one study to address. Nevertheless, the results will provide an initial empirical basis to evaluate reproducibility and may help guide the broader discussion about reproducibility toward areas of significant challenge, productive areas for further inquiry, and possible interventions for improvement. Given the importance of these questions, it is important to recognize what will and will not be learned from the results of the Reproducibility Project: Cancer Biology.

### Does a failure to replicate mean that the original result was a false positive?

No. There are many reasons that two studies of the same phenomenon could obtain different results, and only one of those is that the original was a false positive. The project design minimizes but does not eliminate the possibility of other explanations such as the replication being a false negative due to insufficient power, error in analysis, differences in statistical methods, or error in the design and implementation of the study procedures such as reagent variability/lack of validation, unintentional selective reporting, lack of appropriate controls, lack of equipment calibration, or unrecognized experimental variables (*Ioannidis, 2005*; *Nieuwenhuis et al., 2011*; *Loscalzo, 2012*; *Haibe-Kains et al., 2013*; *Pusztai et al., 2013*; *Vasilevsky et al., 2013*; *Hines et al., 2014*; *Ioannidis et al., 2014*; *Perrin, 2014*).

Other causes for different research outcomes between original and replication have implications for understanding the phenomenon itself. For instance, the original effect may be real but overestimated by the original study. As such, replication may provide new insight, not of the truth of the effect, but of its practical implications. Another reason for different outcomes is that the conditions necessary to obtain the result are not yet understood. Together the replication protocol and peer review of the Registered Report are intended to produce an experimental design for which there is no reason to expect a priori a different result than the original. Those expectations are based on the present theoretical understanding of how and why the effect occurs. However, that understanding may be incorrect or incomplete. Particular features of the original experimental protocol might be critical but unidentified. Therefore, if there is no reason to expect a different result, and a different result is obtained, then differences between the original and replication deemed previously to be non-consequential are now targets for hypothesizing and investigation. This may produce new discoveries and enhance understanding of the effect, the conditions necessary to obtain it, and its implications for biology. This type of discovery is unlikely to occur without direct replication.

### Does a successful replication mean that the original interpretation is correct?

No. Successful direct replication can verify that a result can be obtained and establishes some generalizability by showing that it can be obtained in different circumstances. However, direct replication does not confirm the interpretation of the result. For example, if an original design has an unidentified confounding influence, then the direct replication is also likely to be influenced by that confound.

Developing understanding for the meaning of research findings is often clarified more productively through conceptual replication (*Schmidt, 2009*). In direct replication the original methodology is reproduced as faithfully as possible; in conceptual replication the original research question is tested again with different methods. Conceptual replication can include changes to the model system used, leveraging a new technology or improved procedure, or an operational change to the manipulation or measurement of critical variables. Such changes are done to remove alternative explanations and demonstrate that the phenomenon is not idiosyncratic to the original procedures. Conceptual replication is as vital for

gaining understanding of an effect as direct replication is for increasing confidence that the effect occurs.

## Conclusion

Replication is central to the progress of science: if others cannot reproduce the evidence backing a scientific claim, then the claim loses status as scientific knowledge. This process differentiates science from other ways of knowing for which the power, authority, ideology, or persuasiveness of the person making the claim determines its truth.

The Reproducibility Project: Cancer Biology uses an open methodology to examine reproducibility in cancer biology research. The implications of the project may depend on its outcomes. A high rate of reproducibility might affirm current research and reporting practices, which may suggest that the potentially dysfunctional incentives in the present ecosystem are relatively inert (See *Ioannidis et al., 2014* for a review). On the other hand, a low rate of reproducibility might foster changes by researchers, scientific societies, universities, publishers, and funding agencies to improve research practices and to adjust the training and incentives that maintain them (*Ioannidis and Khoury, 2011*; *Landis et al., 2012*; *Miguel et al., 2014*; *Nosek et al., 2012*; *Wagenmakers et al., 2012*; *Wadmann, 2013*; *Alberts et al., 2014*; *Asendorpf et al., 2013*; *Collins and Tabak, 2014*).

Self-examination is not without challenge. A low reproducibility estimate might prompt concern that the reputation of cancer biology research will be damaged (*Bissell, 2013*). However, we believe that there is much greater risk in having a low reproducibility rate and failing to discover it. Science can only self-correct if there is awareness of what needs correcting. If reproducibility is much lower than expected, then the generation of new knowledge will suffer because it is difficult to pursue innovation and discovery if the foundation of evidence is not reliable (*Forscher, 1963*). A culture that values and practices reproducible science can push out the boundaries of knowledge with confidence that new discoveries have potential to lead to new knowledge and, in the case of cancer biology, cures to one of the greatest challenges to human health.

## Acknowledgements

We would like to thank the following companies for generously donating reagents to the Reproducibility Project: Cancer Biology; BioLegend, Charles River Laboratories, Corning Incorporated, DDC Medical, EMD Millipore, Harlan Laboratories, LI-COR Biosciences, Mirus Bio, Novus Biologicals, and Sigma–Aldrich.

**Timothy M Errington** Center for Open Science, Charlottesville, United States

http://orcid.org/0000-0002-4959-5143

**Elizabeth Iorns** Science Exchange, Palo Alto, United States

**Fraser Elisabeth Tan** Science Exchange, Palo Alto, United States

**Joelle Lomax** Science Exchange, Palo Alto, United States

**William Gunn** Mendeley, London, United Kingdom

**Brian A Nosek** University of Virginia, Charlottesville, United States; Center for Open Science, Charlottesville, United States

## Author contributions

TME, EI, WG, FET, JL, BAN, Conception and design, Drafting or revising the article

*Competing interests:* EI, FET, JL: Employed by and hold shares in Science Exchange Inc. The other authors declare that no competing interests exist.

## Funding

| Funder | Author |
| --- | --- |
| Laura and John Arnold Foundation | Timothy M Errington, Elizabeth Iorns, William Gunn, Fraser Elisabeth Tan, Joelle Lomax, Brian A Nosek |

The Reproducibility Project: Cancer Biology is funded by the Laura and John Arnold Foundation, provided to the Center for Open Science in collaboration with Science Exchange. The funder had no role in study design or the decision to submit the work for publication.

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
