## [Decision Letter]

Thank you for sending your work entitled “An open investigation of the reproducibility of cancer biology research” for consideration as a Feature article at *eLife*. Your article has been favorably evaluated by Sean Morrison (Senior editor), 3 reviewers from our Senior editors and Board of Reviewing Editors (including Charles Sawyers and Joan Massague), and our Features editor.

The Senior editor and the reviewers discussed their comments before we reached this decision, and the Senior editor has assembled the following comments to help you prepare a revised submission.

The reviewers were supportive of this project and believe that this introduction to the reproducibility project will be helpful for the readership of *eLife*. We would ask the authors to consider the following points, and to make minor revisions to the text of the article to address these issues.

1) Why the cancer field was chosen for the replication project should be made clear at the outset. Is this field more prone to publication of data that are not reproducible? One assumes that this choice was made because of the recent pharma reports in which they claimed that they could only replicate ∼20% of a selected set of papers that reported new cancer drug targets. As Errington et al. point out, the data on which the pharma claims were based were not reported and it is therefore hard to evaluate them, and, as others have said, pharma is increasingly relying on NIH-funded academic labs to do the basic research to uncover new drug targets, instead of carrying out target discovery in their own groups, when they would not have this issue!

2) How many of the 50 selected cancer papers have in effect been “replicated” by subsequent publications on the same topic? Would this analysis require a more sophisticated analysis by an expert in the field? Of course, it is also important to note that “me too” studies commonly claim to replicate high impact observations as a way of publishing follow-on observations. Therefore, skepticism is warranted about claims of replication. Nonetheless, it would be worthwhile to indicate what fraction of the 50 papers have ostensibly been “replicated” in studies that claimed to directly test a major conclusion. We would appreciate the authors’ comments on whether it would be possible to include a table showing these data. Feel free to discuss with Sean Morrison if that would be helpful.

3) It is true that exact replication is rarely carried out, but it is unfair to say that innovation is required for publication. Perhaps, in the so-called high profile journals innovation is a criterion, but for most journals in order to be published papers have to report an advance in our understanding of a subject but do not have to be innovative in the normal sense of the word.

4) It would be worth pointing out that in most cases US federal agencies will not award grants for basic research proposals whose goal is simply to replicate, and therefore it would be hard to fund such work even if one wanted to do so.

5) The discussion about “power” is confusing (i.e. 21% power to detect a positive result). Perhaps the authors can provide an example (real or hypothetical) to make this point. Currently it is too abstract for most readers.

6) How many of the citations that were tallied in selecting the 50 highly cited papers for replication were in primary research articles as opposed to reviews? Arguably, reviews should have been discounted, and this might have generated a different set of “high impact” papers to replicate.

7) In the spirit of openness, please include a link to a spreadsheet or other file that includes the 50 papers and their citation rates and altmetric scores.

8) With regard to the discussion about the need for power factors, molecular biology experiments (e.g. sequencing, proteomics, pulldowns) often do not have a real power factor or quantification, and it is not clear how the validity of such data will be evaluated (see next point).

9) The one situation where results should have been replicated is in the lab that published the initial report, and there should be some mechanism for asking the labs how many times a particular experiment was repeated by different lab members. In some papers, one sees statements in the figure legends indicating how many times an experiment was done with similar results (sometimes in response to a reviewer), and in figures where statistical analysis is used this is usually accompanied by an indication of whether the replicates were obtained within the same experiment or between experiments.

10) While it is true that CROs and core services may not have a bias regarding what results to expect, they can or will know what was claimed in the paper describing the experiments they are replicating, which could introduce a hidden bias. Moreover, there is a concern about whether core facilities or CROs necessarily have the expertise to reproduce claims based on innovative techniques, potentially techniques developed in the laboratories that published the original claims. This concern does not necessarily undermine the overall effort, but is worth explicitly addressing.

11) The rationale for selecting the papers is well described and defensible, but it may be worth considering the fact that there are always some highly visible papers published in every field where others have trouble reproducing the work but they never publish the failure to replicate. Perhaps these papers will be chosen by the current method but it may be useful to survey a field for papers that investigators would like to see replicated. I suspect there would be a few that rise up to the top and have great impact on the field if not reproduced. That does not need to be done in the context of the current article, but perhaps worth noting that different approaches could be used in future to identify the controversial articles containing ideas that could be blocking the development of fields.

---

## [Author Response]

*1) Why the cancer field was chosen for the replication project should be made clear at the outset. Is this field more prone to publication of data that are not reproducible? One assumes that this choice was made because of the recent pharma reports in which they claimed that they could only replicate ∼20% of a selected set of papers that reported new cancer drug targets. As Errington et al. point out, the data on which the pharma claims were based were not reported and it is therefore hard to evaluate them, and, as others have said, pharma is increasingly relying on NIH-funded academic labs to do the basic research to uncover new drug targets, instead of carrying out target discovery in their own groups*, *when they would not have this issue!*

We added the following to clarify why we selected cancer biology for investigation: “We selected cancer biology as the area of focus because of the Bayer and Amgen reports, and because of the direct importance of efficient progress in this research discipline for the treatment of disease.”

*2) How many of the 50 selected cancer papers have in effect been “replicated” by subsequent publications on the same topic? Would this analysis require a more sophisticated analysis by an expert in the field? Of course, it is also important to note that “me too” studies commonly claim to replicate high impact observations as a way of publishing follow-on observations. Therefore, skepticism is warranted about claims of replication. Nonetheless, it would be worthwhile to indicate what fraction of the 50 papers have ostensibly been “replicated” in studies that claimed to directly test a major conclusion. We would appreciate the authors’ comments on whether it would be possible to include a table showing these data. Feel free to discuss with Sean Morrison if that would be helpful*.

We agree that this is an important question. For each of the replication papers, preparation of the Registered Report proposal includes a review of the literature for existing replications and the impact of the original findings on further research. This review is time-intensive and will not be completed until all of the Registered Reports have been prepared and submitted to *eLife*. As such, the suggested table will not be available until the final report of the Reproducibility Project: Cancer Biology in which we will summarize the aggregate results across all the studies. We will include the suggested table in that final report. We have added the following sentence: “The replication team will also conduct a literature review for evidence of existing replications although, as noted above, direct replication is likely to be rare. Existing published evidence for replication might indicate the likelihood of reproducing the original results.”

*3) It is true that exact replication is rarely carried out, but it is unfair to say that innovation is required for publication. Perhaps, in the so-called high profile journals innovation is a criterion, but for most journals in order to be published papers have to report an advance in our understanding of a subject but do not have to be innovative in the normal sense of the word*.

We modified the text to address this point. It now reads, “Incentives for scientific achievement prioritize innovation over replication (1; 33). Peer review tends to favor manuscripts that contain new findings over those that improve our understanding of a previously published finding. Moreover, careers are made by producing exciting new results at the frontiers of knowledge, not by verifying prior discoveries.”

*4) It would be worth pointing out that in most cases US federal agencies will not award grants for basic research proposals whose goal is simply to replicate, and therefore it would be hard to fund such work even if one wanted to do so*.

Agreed. That has been added with the following: “These challenges are compounded by the lack of funding support available from agencies and foundations to support replication research.”

*5) The discussion about “power” is confusing (i.e. 21% power to detect a positive result). Perhaps the authors can provide an example (real or hypothetical) to make this point. Currently it is too abstract for most readers*.

We added explanation of the meaning of power and its implications for understanding the positive result rate. The text now reads: “In neuroscience, for example, Button et al. observed the median power of studies to be 21% (6), which means that assuming the finding being investigated is true and accurately estimated, then only 21 of every 100 studies investigating that effect would detect statistically significant evidence for the effect. […] This discrepancy between observed power and observed positive results is not statistically possible. Instead, it suggests systematic exclusion of negative results (15) and possibly the exaggeration of positive results by employing flexibility in analytic and reporting practices that inflate the likelihood of false positives (42).”

*6) How many of the citations that were tallied in selecting the 50 highly cited papers for replication were in primary research articles as opposed to reviews? Arguably, reviews should have been discounted, and this might have generated a different set of “high impact” papers to replicate*.

We used total citation count from the various sources. Thus, some of the citing articles could be reviews. It would be interesting to know how much the list would differ if review citations were excluded, but with present search technologies it would be very laborious to assess that at the present scale. We clarify the citation counting process, noting: “Citations were counted from all sources, which include primary research articles and reviews.”

*7) In the spirit of openness, please include a link to a spreadsheet or other file that includes the 50 papers and their citation rates and altmetric scores*.

Yes, that spreadsheet is available on the OSF project page (http://osf.io/e81xl/studies), and we have added a link to the file in the manuscript: “Details on the selection process and a list of the selected and excluded papers are available at the Open Science Framework.”

*8) With regard to the discussion about the need for power factors, molecular biology experiments (e.g. sequencing, proteomics, pulldowns) often do not have a real power factor or quantification, and it is not clear how the validity of such data will be evaluated (see next point)*.

We agree that this is a point worth clarifying. Sequencing and proteomics studies and experiments are excluded from this investigation because of the challenges identified by the reviewers. We added a note on this: “Also, articles reporting sequencing results, such as publications from The Cancer Genome Atlas project, were excluded. However, if sequencing or proteomic experiments were only part of an article, the other experiments in those papers could still be eligible.”

It is true that pulldowns are often not quantified and usually evaluated as present/not present. However, they can be quantified and usually generate very large effect sizes. We will address pulldown studies that do not have inferential tests or effect sizes in the original reports by following the approach described in the text.

*9) The one situation where results should have been replicated is in the lab that published the initial report, and there should be some mechanism for asking the labs how many times a particular experiment was repeated by different lab members. In some papers, one sees statements in the figure legends indicating how many times an experiment was done with similar results (sometimes in response to a reviewer), and in figures where statistical analysis is used this is usually accompanied by an indication of whether the replicates were obtained within the same experiment or between experiments*.

Agreed. We have added a note that we will investigate whether there are additional replications available that are not noted in the original article: “In these cases, we will inquire with the original authors if there are additional unpublished replications, if it is not already stated in the article, and if any details are available about the results.”

*10) While it is true that CROs and core services may not have a bias regarding what results to expect, they can or will know what was claimed in the paper describing the experiments they are replicating, which could introduce a hidden bias. Moreover, there is a concern about whether core facilities or CROs necessarily have the expertise to reproduce claims based on innovative techniques, potentially techniques developed in the laboratories that published the original claims. This concern does not necessarily undermine the overall effort, but is worth explicitly addressing*.

We agree that our original statement was too strong. For the first part of this comment, we revised our statement to the following: “An advantage of these labs – commercial contract research organizations (CROs) and core facilities – is that they are less likely to be biased for or against replicating the effect. This may reduce the effect of experimenter expectations on observed results (40). However, it does not necessarily eliminate expectancy effects as the replicating researchers are aware of the original findings, and they may have expectations about whether the same result is likely to be obtained or not.”

For the second part of this comment, we added the following:

“Articles were removed if they were clinical trials, case studies, reviews, or if they required specialized samples, techniques, or equipment that would be difficult or impossible to obtain.”

Also: “Next, a key part of matching experiments with laboratories is to identify labs with the appropriate expertise to maximize research quality. This is particularly important with new and innovative techniques, though most techniques called for in the selected experiments are standard techniques for which expertise is widely available. As experiments are matched to labs, it is possible that no appropriate service provider can be identified. If appropriate expertise is not available, then the finding or paper will be excluded from the project.”

*11) The rationale for selecting the papers is well described and defensible, but it may be worth considering the fact that there are always some highly visible papers published in every field where others have trouble reproducing the work but they never publish the failure to replicate. Perhaps these papers will be chosen by the current method but it may be useful to survey a field for papers that investigators would like to see replicated. I suspect there would be a few that rise up to the top and have great impact on the field if not reproduced. That does not need to be done in the context of the current article, but perhaps worth noting that different approaches could be used in future to identify the controversial articles containing ideas that could be blocking the development of fields*.

We agree. Our selection method is one of many reasonable approaches. We added a note of possible alternatives for future research with the following comment: “There are a variety of alternative sampling strategies that could be pursued in parallel efforts such as community nomination of findings that are important to replicate, request for authors to have their published findings replicated (e.g., the Reproducibility Initiative project), or selection of a sample from a particular journal or on a specific topic for focused investigation.”